# The Role of Nutrition in Pathogenesis of Uterine Fibroids

**DOI:** 10.3390/nu15234984

**Published:** 2023-12-01

**Authors:** Jarosław Krzyżanowski, Tomasz Paszkowski, Sławomir Woźniak

**Affiliations:** 3rd Chair and Department of Gynecology, Medical University of Lublin, 20-094 Lublin, Poland

**Keywords:** leiomyoma, nutrients, diet, vitamins

## Abstract

Uterine fibroids are benign tumors that arise from the smooth muscle tissue of the uterus and are the most common tumors in women. Due to their high prevalence, costs for the health care system and the substantial impact on women’s quality of life, they are a significant public health concern. Previous literature on the impact of diet on the occurrence, growth and symptoms of fibroids is limited. Recently, many papers have been written on this topic. A scoping review of PubMed and Cochrane databases was performed using the following keywords: uterine fibroids, antioxidants, diet, diet, vegetarian, vegetables, fruits, meat and soy foods, dairy products, tea, vitamin D, vitamin C, ascorbic acid. Preliminary research has shown a beneficial effect of vegetable and fruit consumption on the occurrence of fibroids. A relationship between hypovitaminosis D and an increased risk of fibroids has also been demonstrated. Studies on epigallocatechin gallate showed its apoptosis-promoting and antifibrinolytic effect in fibroid cells. Initial results are promising, but further randomized trials are needed to draw firm conclusions about the effects of diet and nutrients on uterine fibroids.

## 1. Introduction

Uterine fibroids (UF), also known as leiomyomas, are benign tumors originating from smooth muscle tissue of the uterus. They consist of a large amount of extracellular substance containing fibronectin, collagen and proteoglycans. Fibroids are surrounded by a pseudocapsule, which includes compressed collagen, muscle fibers, blood vessels and neurofibers. Their prevalence increases with age, peaking in the fourth and fifth decades of life. It is influenced by factors such as ethnicity, family history, and hormonal exposure. Being estrogen-dependent tumors, their occurrence before menarche is rare and their size generally decreases after the menopause. Accurately determining the prevalence of fibroids in the population is challenging, as a significant number remain asymptomatic and remain undiagnosed. Depending on the population, studies indicate that 5.4% to 77% of patients have fibroids. In about 30% of cases leiomyomas are symptomatic. Risk factors for UF include age, low parity, early menarche, diabetes, obesity, polycystic ovary syndrome (PCOS) and African ethnicity [1]. A number of factors may explain increased incidence of UF in black women, including higher serum estrogen levels and a higher risk of being overweight or obese compared to white women. In addition, differences in gene expression, higher exposure to chronic stress, poor diet, frequent use of hair straighteners as well as environmental and occupational exposures might also affect the occurrence of UF [2,3,4].

Uterine fibroids represent a significant public health concern due to their high prevalence, substantial costs for the health care system and their impact on women’s quality of life [5]. Although many women remain asymptomatic, for other patients, fibroids can lead to chronic pain, heavy menstrual bleeding, anemia and reproductive difficulties. Fibroids are associated with approximately 10% of infertility cases. In 1–3% of cases, they are the exclusive cause of infertility [1]. Women suffering from fibroid-related symptoms often report a reduced quality of life including emotional, physical and social consequences [6]. Management options range from regular check-ups with gynecologist in asymptomatic cases to surgical interventions such as myomectomy or hysterectomy. Hysterectomy is considered as definitive treatment for uterine fibroids; however, with the increasing number of women wishing to avoid surgery and/or preserve their uterus, minimally invasive and medical treatments are being developed. Among minimally invasive procedures, uterine artery embolization is the most widely recognized treatment. While ultrasound or magnetic-resonance-guided high-intensity focused ultrasound is still considered as experimental method, preliminary reports are promising. In terms of medical therapies, it is possible to distinguish between hormonal and non-hormonal treatments. Hormonal therapies include the levonorgestrel intrauterine system (IUD-LNG), gonadotropin-releasing hormone (GnRH) agonists, GnRH antagonists, combined oral contraceptives and selective progesterone receptor modulators. Non-hormonal therapies include tranexamic acid and non-steroidal anti-inflammatory drugs [1]. Additionally, emerging therapies, including vitamin D supplementation and green tea extracts, have been investigated for their potential role in fibroid management [7,8,9]. The decision regarding treatment method depends on patient’s age, symptoms, procreation plans and other health conditions. Pharmacological treatment is an important option for patients who are either ineligible or unwilling to undergo invasive treatment [1]. Some patients, aiming to alleviate their symptoms while avoiding hormonal or surgical treatment, resort to complementary and alternative medicine options such as exercise, herbs, diet, acupuncture and physical therapy with varying effectiveness [10,11].

In the last century, the most common cause of death shifted from infectious or communicable diseases to chronic, non-communicable diseases. These include, among others, obesity, cerebrovascular disease, cancer and diabetes mellitus. Diet plays a crucial role in the prevention and development of many of these diseases. Adopting healthy eating not only extends life but also enhances the quality of those additional years [12]. The effect of diet on uterine fibroids has been a subject of speculation for many years. However, the lack of high-quality research prevents drawing definitive conclusions [13]. Most studies on nutrition and uterine fibroids concentrate on dietary components such as vegetables, fruits, carotenoids, soy-derived products or vitamin D and their impact on the prevalence of UFs. Some research has also explored the impact of other vitamins, cereal, dairy, green tea or environmental pollutants.

Due to limited knowledge on molecular mechanisms, there is a lack of long-term non-invasive treatment options for UF [14]. Moreover, in most cases, the impact of diet and nutrition on the risk and progression of uterine fibroids remains unclear, with existing data being insufficient to draw firm conclusions. Given the diversity of research regarding nutrients such as vegetables and fruits or vitamin D and, on the other hand, the sparse data and limited studies regarding other nutrients and vitamins, we have undertaken a scoping review. The aim of this review is to investigate the effect of dietary components and interventions in pathogenesis, prevalence and symptoms of uterine fibroids. Identifying potential gaps in the existing literature and highlighting the mechanisms by which nutrition influences the development of UF may provide directions for future research. 

## 2. Methodology

According to the Preferred Reporting Items for Systematic reviews and Meta-Analyses (PRISMA) extension for Scoping Reviews we conducted a thorough review of the literature was utilizing PubMed and Cochrane databases. Keyword combinations included “uterine fibroids” and “antioxidants”, “diet”, “diet, vegetarian”, “vegetables”, “fruits”, “meat”, “soy foods”, “dairy products”, “tea”, “vitamin D”, “vitamin C”, “ascorbic acid”, “Vitamin E, “vitamin A”, “vitamin B complex”, “carotenoids”, “selenium”, “epigallocatechin gallate”. Records in English and Polish published between January 2010 and November 2023 in peer-review journals were included. Exclusions were records written in languages other than Polish and English, case reports case series, studies not addressing nutritional aspects, papers focusing on other types of tumors or gynecological diseases and records lacking an abstract. In a two-stage screening process, titles and abstracts were initially assessed, followed by an analysis of full-text articles [Figure 1]. We extracted relevant information, including study design, sample size, treatment interventions and outcomes from the selected articles and then synthesized it [15]. To complement this search, we scanned the references of relevant reviews. The findings are presented narratively and are supplemented by a table summarizing the main study features and conclusions [Table 1].

## 3. Vegetables and Fruits

Several studies indicate a correlation between a diet rich in vegetables and fruits and the risk for uterine fibroids. Phytochemicals, which are compounds found in plants, possess disease-preventive properties and also contribute to plant color. Phytochemicals are abundant in fruits, vegetables, grains, beans, nuts and seeds. Phytochemicals include, among others, flavonoids, carotenoids and polyphenols. These compounds are known for their ability to regulate cell proliferation, inflammation, fibrosis, apoptosis and angiogenesis [34]. In an in vitro study on leyomyoma cells, methanolic extract of two strawberry strains rich in phytochemicals Romina (R) and Alba (A), as well as Romina anthocyanin (RA) extract, were administered for 48 h. UF cells treated with R and RA exhibited significant decreases in collagen, fibronectin, activin A and versican expression. RA was also the most efficient in inducing apoptosis of UF cells [35]. In the Black Women’s Health Study (BWHS) including 59,000 African American women, 6627 were diagnosed with uterine fibroids. Based on the information obtained from patients, an inverse correlation was found between more servings of vegetables and fruits, including citrus, per day and the occurrence of fibroids. However, no significant relationship was observed between the intake of vitamins C, E, folic acid, carotenoids or fiber and uterine leiomyomas [16]. In a prospective cohort study by Davis et al., it was concluded that participants with the highest intake of fruits had a lower risk of uterine fibroids than those consuming the least amount of fruits, also suggesting an inverse correlation between amount of pesticide residues on vegetables and fruits and the prevalence of uterine fibroids [17]. A Chinese study investigating the influence of a vegetarian diet on the prevalence of fibroids found that consumption of broccoli, cabbage, tomato, Chinese cabbage and apple significantly reduced the prevalence of leiomyomas [18]. Similar results regarding vegetable and fruit intake were achieved in other studies conducted on the Chinese population [8,19]. Given these research outcomes, as well as the overall positive health impact of eating more vegetables and fruits, it is sensible to advise patients at risk or with uterine fibroids to increase their intake of these nutrients. 

## 4. Dairy

Dairy is raw or processed food products made from milk. Dairy products contain many minerals and vitamins, including calcium, magnesium and vitamin D, which may be responsible for their effects in inhibiting tumorigenesis and inflammation [20]. Wise et al. in a prospective study on black women, observed that a higher diary intake reduces the risk of uterine fibroids [21]. Orta et al. concluded that consumption of yogurt and higher calcium intake reduced the risk of fibroids. However, the effect of dairy products in general on the occurrence of fibroids was unclear [20]. In contrast, a prospective cohort study on Chinese population by Gao et al. indicated that increased consumption of milk and soybean elevated the risk of uterine fibroids. It is important to note, however, that this study did not distinguish between milk and soybean consumption [22]. Given the limited data available, further research is needed to clarify the effect of dairy products on uterine fibroids. 

## 5. Soybean

Soybean and soy-derived foods are key source of protein for many people across the world [36]. Rich in daidzein and genistein, soybeans contain isoflavones, which are nonsteroidal phenolic plant compounds and are among the most estrogenic substances. These isoflavones belong to the group of phytoestrogens with structural similarities to 17β-estradiol [37]. It was hypothesized that high consumption of foods containing soy protein may have an influence on the endocrine system, as well as on the prevalence and growth of fibroids. However, the existing data on this topic remain limited [36]. Two studies have investigated the correlation between urinary isoflavone levels and the risk of uterine fibroids. In the first study, patients with UF exhibited elevated urine enterolactone levels, but there was no observed association between urinary genistein, daidzein, equol (a metabolic product of intestinal bacteria), total isoflavones or phytoestrogens and uterine fibroids [24]. Conversely, in a recent study on US women, Yang et al. identified a positive association between the concentration of isoflavone metabolites in urine and uterine fibroids, with equol demonstrating the most significant effect [9]. Qin et al. performed a meta-analysis to evaluate the influence of soy-derived products on the prevalence of uterine fibroids. This analysis encompassed studies on both the consumption of soy products in infants as well as in adult women. Based on infant studies, soy formula was found to increase the risk of fibroids by 35%. Among adults, consumption of large amounts of soy products was linked to a 92% increase in the UF risk [38]. It is worth noting, however, that this relationship was not the same in all studies. Upson et al. found that although the prevalence of uterine fibroids did not increase in women fed with soy formula during their infancy, the diameter of fibroids was on average 32% higher in these patients [23]. Similarly, Wise et al. concluded with no correlation between soy intake and UF risk [21]. 

## 6. Green Tea Extracts 

Green tea is one of the most popular beverages in the world. It originated in China, spread all over the world and, currently, it is one of the most popular beverages [39]. Green tea is made from the Camellia sinensis plant. It contains significant amounts of catechins like epigallocatechin gallate (EGCG), epigallocatechin and epicatechin gallate, with EGCG being the main antioxidant. It has anti-angiogenic and anti-proliferative effects. EGCG has been studied as a potential treatment option for many diseases, including those affecting the female reproductive system. Several studies have assessed its impact on UFs both in vitro and in vivo. EGCG has an apoptosis-promoting effect in both cell lines and animal models, which resulted in a reduction in the number and size of the observed fibroids [40]. Cyclin D1, a protein involved in cell-cycle progression, Was shown to be increased in UF cells. However, EGCG has been found to reduce these levels. EGCG also appears to inhibit the production of collagen and fibronectin in UF cells [41]. In a study by Roshdy et al., patients received 800 mg of green tea extract (45% EGCG), resulting in a significant reduction in symptoms and the size of UF compared to the placebo group [25]. In a subsequent 6-month observational study by Biro et al., participants taking EGCG-enriched green tea extract capsules (390 mg EGCG daily) reported significant improvement in their physical quality of life. However, no significant change in myoma size and number was detected [26]. In another study on women of reproductive age, no drug-induced liver injury was found during one-month-long therapy with 720 mg ECGC daily [42]. If the preliminary results are confirmed, EGCG may be a viable treatment option for uterine fibroids. 

## 7. Selenium

Selenium intake is linked to a reduced risk of several cancers. It has antioxidant properties and modulates heat shock protein 70. The influence of selenium on uterine fibroids was investigated in a study on Japanese quail. The number of fibroids did not differ between the study (selenium supplementation) and control groups; however, birds in study group had smaller fibroids [43]. Additional research is necessary to support these findings in both animals and humans. 

## 8. Curcumin

Curcumin, a polyphenol extracted from Curcuma longa, exhibits an inhibitory effect on the proliferation of many tumor cell lines. However, its exact mechanism of action is unknown. A study on mice with human leiomyoma xenograft demonstrated an inhibitory effect of curcumin on the growth of xenografts, as well as a reduction in the production of matrix proteins. An in vitro study on Eker rat-derived uterine leiomyoma cell lines showed that curcumin significantly inhibits cell proliferation [44,45]. Extensive studies on animals and humans are necessary to substantiate the preliminary results regarding the effect of curcumin on UF.

## 9. Cereal

The consumption of cereals seems to have no impact on the occurrence of uterine fibroids or their symptoms; however, additional research in this area is required to confirm these results [19,46]. 

## 10. Vitamins

Vitamins are essential dietary components that are not synthesized in the human body, or are synthesized in quantities insufficient to maintain health. The daily requirements of most vitamins have been well established. Their deficiency leads to the development of various diseases, including scurvy, anemia, pellagra and osteoporosis [47,48,49]. The most widely studied vitamin in the context of uterine fibroids is vitamin D. 

### 10.1. Vitamin D

Vitamin D is a group of steroid compounds including calciferol-D1, ergocalciferol-D2 and cholecalciferol-D3. It is involved in cell cycle regulation, cell differentiation and calcium–phosphate balance [50]. Produced in the human body through exposure to sunlight, it can also be sourced from the diet and supplements. The nutrients containing the most vitamin D include fatty fish and fortified foods. Small quantities of vitamin D can be found in foods such as cheese, egg yolks or beef liver [39]. Vitamin D seems to inhibit the proliferation of fibroid cells, while its deficiency may induce inflammation within the myometrium [20]. Through its nuclear receptor (vitamin D receptor—VDR) and activation of tyrosine kinase, it affects various signaling pathways. Immunohistochemistry studies showed reduced amounts of VDRs in fibroids compared to the myometrium surrounding the UF and reduced amounts of VDRs in the vicinity of the fibroid compared to the normal myometrium [51]. CYP24A1 is responsible for coding 24-hydroxylase, a vitamin D3-catabolizing-enzyme. Its expression in fibroids is higher than in normal myometrium and it leads to relative hypovitaminosis D inside the UF [52]. Meanwhile, data presented by Ali et al. showed that vitamin D3 treatment mitigates pathogenic DNA damage and may explain the beneficial effects of vitamin D3 in studies on rats and human UF cells and patients [53]. Ciebiera et al. found that serum vitamin D levels are significantly lower in patients with UF. The authors concluded that vitamin D deficiency may be a risk factor for UF [27]. Similar results were achieved in the Chinese population [28]. In a recent study by Harmon et al., 25-hydroxyvitamin D concentrations in blood serum above 20 ng/mL were associated with reduction in UF growth, while concentrations above 30 ng/mL were associated with decreased incidence of UF [30]. In a 12-month follow-up, patients receiving Vitamin D supplementation experienced a decrease in the UF volume. In addition, patients in this group required fewer surgical interventions [29]. In a study by Grandi et al., significant reduction in the size of UF was observed after three months of combined treatment with vitamin D 50 µg, EGCG 300 mg and Vitamin B6 10 mg [54]. In conclusion, given the prevalence of Vitamin D deficiency around the world [55] and the health benefits of its supplementation, in the future, vitamin D may be considered as a method of preventing or slowing down the development of uterine fibroids. However, randomized trials are still needed to determine the precise effect of vitamin D on the pathogenesis, prevalence and size of uterine fibroids. Determining the appropriate dose and target serum vitamin D level in patients with uterine fibroids also requires further research.

### 10.2. Vitamin C

Ascorbic acid, a water-soluble vitamin, was first synthesized in 1923. Its presence in the diet is essential for the physiological functioning of the human body. Vitamin C is involved in the synthesis and metabolism of folic acid, tyrosine and tryptophan. It accelerates the hydroxylation of glycine, lysine and proline by keeping the active center of metal ions in a reduced state. The main sources of vitamin C are fruits and vegetables. Its concentration in food of animal origin is low. A deficiency in vitamin C leads to anemia, infections, scurvy and other health issues. Additionally, some research suggests that ascorbic acid may also influence both male and female infertility [56]. During the literature review, we found limited data on the effect of vitamin C on occurrence, growth and symptoms of uterine fibroids. In the BWHS study, there was no observed correlation between vitamin C intake and the occurrence of fibroids [16]. Two randomized studies investigated the effect of administering two grams of ascorbic acid on perioperative blood loss during myomectomy. In a study by Lee et al., vitamin C had no impact on blood loss during laparoscopic myomectomy. Conversely, in a study by Pourmatroud et al., blood loss during abdominal myomectomy was significantly reduced in the group receiving ascorbic acid [57,58]. Currently, the available data on the effects of vitamin C on fibroids are insufficient to draw any conclusions. 

### 10.3. Vitamin A

Vitamin A, a fat-soluble compound, owes its broad effects in metabolic processes to its β-ionone ring and isoprenoid chain. Food provides vitamin A in the form of retinols and carotenoids. Retinols are typically found in foods of animal origin, while the main sources of carotenoids are vegetables and fruits. Vitamin A is stored in hepatocytes in the ester form and is de-esterified as needed. Retinoid acid is a biologically active form of vitamin A. By binding to nuclear receptors, it enables transcription [59]. Initially, in a BWHS study, Wise et al. concluded that animal-derived vitamin A seems to be inversely associated with uterine fibroids risk in black women [16]. However, in a subsequent study on black women, no association was identified between the intake of lycopene or other carotenoids and UF incidence [31]. Furthermore, Martin et al. demonstrated that women with medium to high levels of vitamin A are at increased risk of uterine fibroids, compared to those with low concentrations of vitamin A [32]. 

### 10.4. Vitamin E

Vitamin E encompasses a group of compounds, including tocopherols and tocotrienols. α-tocopherol, the most active and second most common form of vitamin E in diet, was discovered over a century ago [60]. It exhibits antioxidant properties, protects the plasma membrane and regulates gene expression. Because of its structural determinants, it can bind with estrogen receptors [33]. Deficient intake of both vitamin C and E was found in one-third of Latin American women. Vitamin E is associated with infertility, and some studies suggest a higher rate of adverse pregnancy outcomes in patients with its deficiency [61]. Our literature review revealed only a few studies addressing the impact of vitamin E on uterine fibroids. Wise et al., in a BWHS study, reported no correlation between vitamin E intake and the incidence of uterine fibroids [16]. Conversely, Ciebiera et al., in a study on Caucasian women, found that the concentration of alpha tocopherol was significantly lower in women with uterine fibroids [33]. Similarly to the BWHS study, Martin et al. did not find statistically significant association between vitamin E and uterine fibroids [32].

### 10.5. Vitamin B3

Niacin, a member of the B complex vitamins, is a part of a group of water-soluble vitamins that also includes thiamine, riboflavin, pantothenic acid, pyridoxine, biotin, folate and cobalamin. Niacin serves as a precursor of nicotinamide adenine dinucleotide (NAD) and nicotinamide adenine dinucleotide phosphate (NADP) coenzymes, which are essential for DNA repair and cholesterol synthesis [62]. Fletcher et al. observed that uterine fibroid tissue has a different nicotinamide adenine dinucleotide phosphate oxidase (NOX) profile from normal myometrium. This leads to a prooxidant state in fibroids, thereby influencing their development [63]. During our literature review, no studies were identified that directly examined the impact of vitamin B compounds on the occurrence and symptoms of uterine fibroids. In general, the effect of vitamins on UF is most comprehensively described in the case of vitamin D, with most studies supporting its protective effect. For other vitamins, only a few studies have investigated their involvement in the pathogenesis and occurrence of fibroids.

## 11. Strengths and Limitations of Available Studies

There are several notable strengths in the studies analyzed in this review. Firstly, the studies feature diverse population samples with research conducted in different countries and on different populations, though a majority of the papers identified focus predominantly on African American women. Second, several studies provided comprehensive longitudinal data tracking large groups of women over an extended duration. In some studies, the authors employed multivariate analysis to control for potential confounding studies. Despite these strengths, there are some possible limitations. Many of the studies found were based on self-reported data. Furthermore, we found a lack of interventional studies comparing the effects of different nutrients, such as vitamin D therapy vs. vitamin D combined with EGCG.

## 12. Conclusions

Diet plays a crucial role in maintaining health and preventing many diseases. Current research suggests that eating habits generally deemed healthy, such as increased consumption of vegetables and fruits or vitamin D supplementation, might positively impact the occurrence of uterine fibroids. Dietary education should be an element of prophylaxis and treatment in all patients, including those with or at risk of developing UF. Preliminary research on epigallocatechin gallate appears to be promising. If the data regarding its safety and effectiveness are confirmed, EGCG may become an option for the treatment of uterine fibroids. 

In Caucasian and Asian women, while fibroids are less prevalent, they still constitute a significant burden on the health care system and require further research. Large-scale, randomized studies are necessary to better understand the impact of nutrition on the pathogenesis and occurrence of UF in these populations. Conducting comparative studies is important to develop effective methods for the prevention of UF using nutrients or supplements. These studies should examine the optimal dose that achieves therapeutic effect and assess whether it is associated with side effects.

## Figures and Tables

**Figure 1 nutrients-15-04984-f001:**
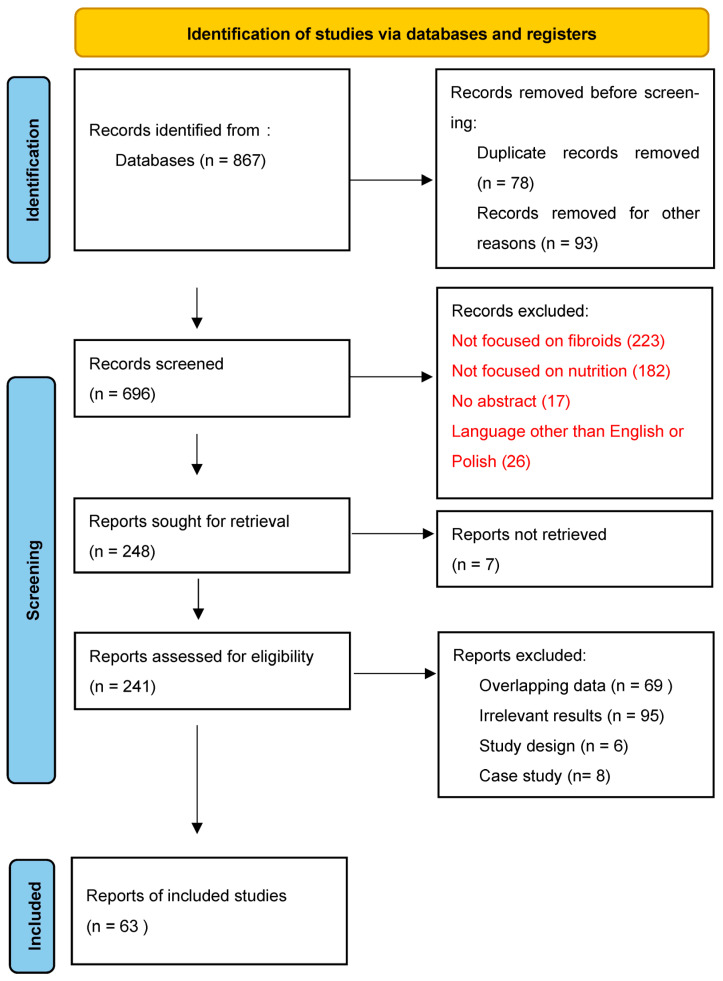
PRISMA 2020 flow diagram [15].

**Table 1 nutrients-15-04984-t001:** Effect of nutrients on uterine fibroids.

	Reference	Group	Country	Outcome
Vegetables and fruits	Zhou, 2020 [8]	248 women; 37 patients with uterine fibroids (UF)	China	Carrot, kiwi and seaweed intake inversely associated with the risk of fibroids
	Wise, 2011 [16]	22,583 women 6627 patients with UF	USA	Inverse association between fruits and vegetables intake and UF
	Davis, 2023 [17]	9706 patients with UFs confirmed by ultrasound or hysterectomy	USA	Lower risk of fibroids in patients with highest intake of fruits Inverse association between amount of pesticide residues and UF risk
	Shen, 2016 [18]	600 Chinese Han women with and 600 without UF	China	Lower intake of broccoli, cabbage, Chinese cabbage, tomato and apple in patients with UF
	He, 2013 [19]	73 women with and 210 without UF	China	Vegetable and fruit intake reduces the risk of UF
Dairy products	Orta, 2020 [20]	81,590 women; 8142 cases of ultrasound or hysterectomy confirmed UF	USA	Inverse correlation between yogurt intake and UF risk
	Wise, 2010 [21]	22,120 women; 5871 cases of UF	USA	Weak inverse correlation of yogurt intake and UF
	Gao, 2018 [22]	213 patients with UFs and 1060 without	China	Frequent milk or soy consumption is a risk factor for UF
Soy foods	Yang, 2023 [9]	1579 participants form the National Health and Nutrition Examination Survey; 204 patients with UF	China	Positive correlation between mixed metabolites of urinary phytoestrogens and UFs, with greatest impact from equol.
	Upson, 2016 [23]	1553 African American women; 345 cases of UF detected with ultrasound	USA	Soy intake during infancy correlated with larger fibroid size
	Simon, 2015 [24]	157 patients with UF and 171 without	Jamaica	Higher urine enterolactone levels in UF patients. No association with increased risk of UF.
	Gao, 2018 [22]	213 patients with UF and 1060 without	China	Frequent milk or soy consumption is a risk factor for UF
	Wise, 2010 [21]	22,120 women; 5871 cases of UF	USA	No association between soy intake and UF
Green tea	Roshdy, 2013 [25]	22 women treated with 800 mg of green tea extract and 11 with placebo	USA	Reduced symptoms and size of fibroids in the study group
	Biro, 2021 [26]	25 women with UF	Germany	Green tea extract associated with higher quality of life. No change in UF size
Cereal	He, 2013 [19]	73 women with and 210 without UF	China	No association between cereal consumption and UF
Vitamin D	Ciebiera, 2016 [27]	105 women with and 83 without UF	Poland	Lower levels of serum 25-hydroxyvitamin D (25OHD) in patients with UF
	Li, 2020 [28]	279 women with and 267 without UF	China	Lower levels of serum 25OHD in patients with UF
	Ciavattini, 2016 [29]	108 patients with UF; 53 underwent vitamin D supplementation (study group)	Italy	Lower surgical or medical treatment rates in study group
	Harmon, 2022 [30]	1610 women without previous diagnosis of UF	USA	Serum 25-OHD > 20 ng/mL associated with reduction in UF growth; >30 ng/mL associated with decreased incidence of UF.
Vitamin C	Wise, 2010, [21]	22,120 women; 5871 cases of UF	USA	No association between vitamin E intake and the incidence of UF
Vitamin A	Wise, 2010 [21]	22,120 women; 5871 cases of UF	USA	Animal derived vitamin A inversely associated with UF risk
	Wise, 2021 [31]	1230 patients; 301 cases of UF diagnosed in 5-year follow up	USA	No association between intake of lycopene or other carotenoids and UF incidence
	Martin, 2011 [32]	887 patients; 68 with self-reported UF	USA	Patients with middle and higher levels of vitamin A are at increased risk of UF
Vitamin E	Wise, 2010 [21]	22,120 women; 5871 cases of UF	USA	No association between vitamin E intake and the incidence of UF
	Martin, 2011 [32]	887 patients; 68 with self-reported UF	USA	No association between vitamin E and UF
	Ciebiera, 2018 [33]	162 patients; 100 cases of UF	Poland	Lower levels of alpha tocopherol in patients with UF
Vitamin B3		No studies on humans found		

## Data Availability

Not applicable.

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
