# Peer review of "The Role of Nutrition in Pathogenesis of Uterine Fibroids"

_nutrients, 2023, doi:10.3390/nu15234984_

Round 1

Reviewer 1 Report

Comments and Suggestions for Authors

The manuscript “The role of nutrition in pathogenesis of uterine fibroids” contributes to giving information about the relationship between nutrition and uterine fibroids related to female infertility risk. The authors described the information based on several studies' findings. However, this manuscript feels like this is too general information. I believe that the accumulation of information such as this manuscript is very important for future clinical applications. Especially, it needs additional schemes of their results to make it easier for the reader to read.

Some points have to be corrected.

Major points

1. In Figure 1, what was the author's data search based on?

2. The results and Discussion parts require some description of the molecular mechanism between nutrition and uterine fibroids because there is too general information in manuscripts.

3. As described in the manuscript, the effect of one factor is known, but are there no reports on the effect of two or more factors combinations.

4. Why are black people more likely to get uterine fibroids? Is the cause environmental factors or lifestyle-specific?  

1. In this study, the authors focused on the association between circulating lipids and female infertility. It is better to add a schematic model of their relationship and how they are related to pregnancy failures.

Minor points

1. Lines 154: Add after “study” comma.

2. Line 155: Amend “then” to “than”.

2. Line 162: Amend “are” to “is”.

3. Line 189: Amend “studies on” to “on studies”.

4. Line 206: Amend “al.” to “al.,”.

5. Line 233: Amend “pellarga” to “pellagra”.

6. Line 242: Amend “at” to “et”.

Comments on the Quality of English Language

Moderate editing of English language required.

Author Response

Thank you for your valuable comments and suggestions regarding our manuscript. We appreciate your time and effort in reviewing our work.

We have made the following corrections and improvements in the revised manuscript:

Major points:

  1. The materials and methods section as well as Fig. 1 have been supplemented with additional information regarding data search.
  2. We added descriptions of the molecular mechanism to our article.
  3. In our review, we identified a study that examined the effect of vitamin D and EGCG treatment on UF. Some studies have analyzed the impact of more than one factor on UF, for example, study 10.1097/MD.0000000000012009 found that soy and dairy products increase the risk of UF, but the impact of soy products was not compared to dairy products separately. However, there is lack of interventional studies comparing the effects of different nutrients on UF
  4. A paragraph has been added to the introduction regarding the increased occurrence of fibroids in African-American women.
  5. “the authors focused on the association between circulating lipids and female infertility”. We would like to clarify that our study did not primarily focus on these associations. Our research aimed at describing the relationship between nutrition and UF.

Minor points: The corrections were made as suggested.

Moreover, the introduction and results sections were divided into smaller paragraphs for easier understanding by the reader.

Regarding the moderate English editing, the manuscript was reviewed by an English native speaker.

We hope that our revisions and clarifications meet your approval, and we look forward to any further suggestions you may have.

Sincerely,

Jaroslaw Krzyzanowski, Tomasz Paszkowski, Slawomir Wozniak

Reviewer 2 Report

Comments and Suggestions for Authors

Esteemed Authors and Editorial Team,

The chosen theme is not new and is better studied by other authors.

There is at least one other review with the same design and superior results.

https://www.ncbi.nlm.nih.gov/pmc/articles/PMC7908561/

Andrea Tinelli, Int. J. Environ. Res. Public Health 2021, 18, 1066 , Uterine Fibroids and Diet= 52 citations.

The mentioned study is from 2021 and has a number of additional elements: it covers a longer period (1990-2020 Vs 2010-2023), it also discusses the diet that stimulates the growth of uterine fibroids (e.g. 3.3. Dietary Fat, Meat and Fish Intake )

The study does not qualify for publishing and my recommandation is to be rejected

Best regards !

Author Response

Thank you for your critical assessment of our manuscript. We appreciate the opportunity to clarify the unique aspects and contributions of our study in the context of existing literature. While we acknowledge that the theme of uterine fibroids and diet has been previously explored, we believe our study offers distinct and valuable insights that complement existing research. Specifically, our study, covering the period from 2010 to 2023, provides a more contemporary analysis of the topic, capturing the latest advancements and perspectives in this field. This is particularly significant given the rapid evolution of nutritional science and its implications for uterine fibroid management. We are open to further enhancing our and would appreciate any specific suggestions you might have in this regard. Thank you once again for your thorough review and valuable feedback. We look forward to the opportunity to improve our manuscript in line with your recommendations.

Sincerely,

Jaroslaw Krzyzanowski, Tomasz Paszkowski, Slawomir Wozniak

Reviewer 3 Report

Comments and Suggestions for Authors

This is an interesting scoping review which needs several revisions.

- The introduction section should be split into 3 paragraphs.

- The authors should report at the end of the introduction the literature gap for which they decided to write this scoping review.

- The main aim of the present review should also be reported at the end of the introduction section.

- In the material and methods section, the authors should add much more information about the recommended methodology used in the case of scoping reviews.

- In the material and methods section and figue 1, the authors report that they performed a systematic review but also reported that this is a scoping review. This is very confusing and should be revised.

-In the results section the paragraphs are too big and the authors should split the into more paragraphs.

- A discussion section is missing.

- The strengths and the limitations of the currently available studies should be reported and emphasized.

- In the conclusion section the authors should reported what research could be performed to the future to cover the literature gap.

- The authos should try to use more updated references.

- The reference style should be revised according to the recommedations of the journal.

Moderate English language editing is recommended.

Comments on the Quality of English Language

Moderate English language editing is recommended.

Author Response

Thank you for your valuable comments and suggestions regarding our manuscript. We appreciate your time and effort in reviewing our work.

The introduction section was divided into smaller paragraphs for easier understanding by the reader. Furthermore, at the end of introduction we included literature gap, main aim of the study, as well as materials and methods section. The materials and methods section was updated with additional information and revised along with  description of Fig. 1 to avoid confusion.  

The results section was divided into smaller paragraphs for easier understanding by the reader. It was also supplemented with the molecular mechanism between nutrition and UF. The discussion of our findings is integrated in the sections that describe the impact of nutrients on UF.

The conclusions section was supplemented with strengths and the limitations of the currently available studies, as well as with suggested directions for further research to cover literature gap.

During the revision, we added several current studies to the manuscript. References were revised according to the style recommended by the journal. Regarding the moderate English editing, the manuscript was reviewed by an English native speaker.

Thank you once again for your valuable feedback, and we look forward to your further suggestions. Sincerely,

Jaroslaw Krzyzanowski, Tomasz Paszkowski, Slawomir Wozniak

Round 2

Reviewer 1 Report

Comments and Suggestions for Authors

I think that the revised manuscript has been fundamentally improved and that it includes the contents requested by the referees and editorial team. 

Comments on the Quality of English Language

Minor editing of English language required.

Author Response

Dear Reviewer,

we thank for the time and effort that you invested into the review of our manuscript, and for your helpful comments and suggestions.

Sincerely

Jarosław Krzyżanowski, Tomasz Paszkowski, Sławomir Woźniak

Reviewer 2 Report

Comments and Suggestions for Authors

Esteemed Editor and author team,

the study can be accepted in prezent form.

Sincerely

Author Response

(The authors gave the same response as above.)

Reviewer 3 Report

Comments and Suggestions for Authors

The manuscript has significantly been improved. Some minor point should be revised.

-Please, create next to the introduction section a separate section entitled as "Methodology". In this new section, the already existed text of lines 89-141 should be included.

-In line 90, please delete the word "use".

- Please, report Table 1 into the text.

- The two paragrpahs concerning the strengths and the limitations should be transferred at the end of the discussion section.

Author Response

Dear Reviewer,

we greatly appreciate your feedback on our manuscript and the constructive suggestions provided.

The following steps were taken to address the minor points:

  • separate “Methodology” section was created;
  • the word “use” in line 90 has been deleted;
  • table 1 was reported at the end of “Methodology” section.
  • Paragraphs concerning strengths and limitations were transferred to the end of the results and discussion section.

Sincerely

Jarosław Krzyżanowski, Tomasz Paszkowski, Sławomir Woźniak